# Vascular Dysfunctions Contribute to the Long-Term Cognitive Deficits Following COVID-19

**DOI:** 10.3390/biology12081106

**Published:** 2023-08-09

**Authors:** Zahra Shabani, Jialing Liu, Hua Su

**Affiliations:** 1Center for Cerebrovascular Research, University of California (San Francisco), San Francisco, CA 94131, USA; zahra.shabaninabikandi@ucsf.edu; 2Department of Anesthesia and Perioperative Care, University of California (San Francisco), San Francisco, CA 94131, USA; 3Department of Neurosurgery, University of California (San Francisco), San Francisco, CA 94131, USA; jialing.liu@ucsf.edu

**Keywords:** long COVID, SARS-CoV-2, endothelial cells, cognitive dysfunction, blood-brain barrier, neuro-inflammation

## Abstract

**Simple Summary:**

Although SARS-CoV-2 primarily affects the upper respiratory system and the lungs, it can also affect vasculature, leading to the impairment of endothelial and multi-organ function. SARS-CoV-2 can trigger the release of pro-inflammatory cytokines, leading to disruption of tight junction proteins between endothelial cells and impairment of the blood-brain barrier (BBB). This process further allows the infiltration of immune cells and other particles into the brain, worsening brain injury. Prolonged neuro-inflammation and disruption of the BBB have been postulated as the potential primary causes of both acute and chronic cognitive dysfunction in COVID-19 patients. This review explores the effects of COVID-19 on vascular dysfunction and consequent cognitive impairment in patients.

**Abstract:**

Severe acute respiratory syndrome coronavirus 2 (SARS-CoV-2) is a single-stranded RNA virus and a member of the corona virus family, primarily affecting the upper respiratory system and the lungs. Like many other respiratory viruses, SARS-CoV-2 can spread to other organ systems. Apart from causing diarrhea, another very common but debilitating complication caused by SARS-CoV-2 is neurological symptoms and cognitive difficulties, which occur in up to two thirds of hospitalized COVID-19 patients and range from shortness of concentration and overall declined cognitive speed to executive or memory function impairment. Neuro-cognitive dysfunction and “brain fog” are frequently present in COVID-19 cases, which can last several months after the infection, leading to disruption of daily life. Cumulative evidence suggests that SARS-CoV-2 affects vasculature in the extra-pulmonary systems directly or indirectly, leading to impairment of endothelial function and even multi-organ damage. The post COVID-19 long-lasting neurocognitive impairments have not been studied fully and their underlying mechanism remains elusive. In this review, we summarize the current understanding of the effects of COVID-19 on vascular dysfunction and how vascular dysfunction leads to cognitive impairment in patients.

## 1. Introduction

In late 2019, a new variant of coronaviruses, SARS-CoV-2, emerged as a cause of novel severe acute respiratory syndrome, which quickly spread around the entire world and became the newest global health concern [1,2]. SARS-CoV-2 is primarily characterized as a respiratory pathogen transmitted via respiratory droplets, although abnormalities of other organs including the brain, heart, kidneys, liver, skeletal muscle, and skin have also been noticed [3]. A meta-analysis study of 24,410 patients with COVID-19 reported fever, cough, fatigue, and hyposmia as the most common symptoms [4]. Compared to common symptoms of impaired olfaction and gustation that are present in 85% of COVID-19 patients, 36.4% of patients exhibited neurological abnormalities, such as dizziness, headache, and weakened consciousness, during hospitalization [3]. A study using murine models indicated that coronavirus could enter the CNS after infection [5], corroborating the neurological symptoms observed clinically. In addition, long-term cognitive dysfunction has been detected in patients who have recovered from COVID-19 infection and patients with persistent infection [6].

Cognition is defined as the conceptual processing of information that is mandatory for acquiring knowledge and expression of a response. This process includes obtaining information (perception), selecting necessary data (attention), and representing (understanding) and retaining information (memory), leading to behavior control (reasoning and judgment) [7]. Fatigue and lasting impairment of memory, decision making, and concentration are the most common neurological injuries reported in non-hospitalized COVID-19 patients. Additionally, numerous patients have described subtle cognitive and behavioral deficits that are difficult to characterize. These symptoms are commonly described as “brain fog” or “mental clouding” [8]. A recent guideline on long-term effects of COVID-19 published by the National Institute of Health and Care Excellence (NICE) suggests using the term “ongoing symptomatic COVID-19” for indications lasting between 4 and 12 weeks after the acute onset and “post-COVID-19-syndrome” for symptoms lasting more than 12 weeks [9]. Evidence from cohort studies implies the relationship between continuing systemic inflammation during SARS-CoV-2 infectivity and subsequent cognitive failure and hippocampal atrophy. Hence, in cognitive decline caused by COVID-19, attention should be paid to the role of inflammation. Through pro-inflammatory cytokines’ release, SARS-CoV-2 can trigger signaling pathways, finally resulting in disrupted tight junction (TJ) proteins and an impaired blood-brain barrier (BBB). This process further allows the infiltration of immune cells and other particles into the brain, worsening brain injury [10,11]. In this review, we discuss the probable role of vascular dysfunction in causing the cognitive deficiency of COVID-19 patients.

## 2. Long-Term Neurological and Cognitive Dysfunction Following COVID-19

During the COVID-19 pandemic, cognitive abnormalities were commonly detected. Long-term cognitive deficiency is a common manifestation of COVID-19, resulting in problems with memory, concentration, and receptive language, executive function difficulties, and fatigue. Psychiatric indications and cognitive impairment may progress and persist months after the infection. Therefore, this condition is called “Long COVID” [12]. Long-term cognitive outcomes can affect patients’ daily living functions, employment, and the ability to return to work, leading to a large disease burden [13]. Several mechanisms are suggested for long-term cognitive impairment, including disruption of the BBB, neuro-inflammation, synaptic dysfunction, disturbed neurotransmitter release, and neuronal loss. Targeting these processes may have treatment potential for long-term cognitive difficulties [14].

Interests of clinical and public health are therefore no longer restricted to the mortality rate and clinical consequences of hospitalized patients but extend to long-term adverse outcomes of COVID-19 disease, even after recovery and discharge from the hospital. There is a growing body of evidence for the persistent neurological and cognitive symptoms after COVID-19 illness. Given the persistent neurological features after SARS-CoV-2 infection, several studies have reported on the onset of neurodegenerative disease in patients without any previous history. Accordingly, 15% of COVID-19 cases showed one or more neurological signs for the first time, including 13% polyneuro/myopathy, 1% Guillain-Barré syndrome, 2% mild encephalopathy, 1% parkinsonism, and 1% ischemic stroke in the 3 months after infection [15]. In addition, a cohort study of 143 patients revealed headache, hyposmia, and myalgia at least in 5% of the population 2 months after SARS-CoV-2 infection [16].

In the context of cognitive deficits, it has been reported that 20–70% of COVID-19 cases display a quantitative or qualitative disturbance of consciousness during the acute phase [17,18]. In one study, the psychopathological and cognitive status of 226 COVID-19 pneumonia survivors were prospectively assessed one- and three-months following hospital discharge. Regardless of clinical and physical severity, 78% of the participants presented poor function in at least one cognitive domain, while 50 to 57% of the sample displayed impaired executive functions and psychomotor coordination [19]. In another case report, cognitive screens from recovered non-hospitalized COVID-19 cases showed persistent neuro-cognitive symptoms, particularly difficulties in working memory and executive function [20]. Furthermore, a follow-up study of 13,001 participants with relatively mild COVID-19 disease reported a significantly higher prevalence of memory problems eight months after the positive SARS-CoV-2 test compared to the control group in favor of the worsening of health even after recovery from infection [21]. In a separate cohort study, only long-term anosmia, ageusia, memory loss, and headache remained in COVID-19 cases after 60 days, yet memory deficit continued in these patients into the third month of disease [22]. Similarly, according to a follow-up study of 18 mild to moderate COVID-19 patients 85 days after recovery, over 75% of patients reported problems in episodic memory, attention, and concentration [23]. In addition, a comparison of cognitive function in SARS-CoV-2 patients to matched controls suggested significant differences in sustained attention [24], executive function and visuospatial processing [25], attention, memory, and language [26].

Interestingly, a case report describes progression of disease from mild common symptoms like myalgia, fatigue, smelling loss, and memory deficit at the beginning to severe outcomes including right-sided weakness, sensory loss, and worsening cognitive impairment. Neuroimaging confirmed an ischemic infarct in the middle cerebral artery of this case [1,27]. In addition, a recent cross-sectional report studied the connection between COVID-19 disease and cognitive signs in 57 hospitalized participants and found a high prevalence of cognitive impairment (>80% of the sample), primarily in the fields of attention and executive function [28].

Among cognitive performances, fatigue, memory, and concentration difficulties were reported as the most common long-lasting symptoms by 36% of symptomatic COVID-19 patients after 4 weeks and mostly observed in women [9,29,30,31].

According to another study, cases with a history of mild symptomatic SARS-CoV-2 infections have more than 18 times higher risk for cognitive deficits than individuals without clinical manifestations of the infection [32]. In addition, a recent longitudinal study showed the important role of COVID-19 severity in modulating the long-term cognitive outcomes of the disease [33]. Therefore, cognitive changes can occur even after mild COVID-19 illness. However, hospitalized and severely affected patients are at higher risk for persistent cognitive and neurological dysfunction. Moreover, amongst diverse cognitive domains, impairment of episodic and working memory, attention and concentration deficits, consciousness, and executive function difficulty are the major cognitive complications in patients suffering COVID-19 infection. 

## 3. Molecules Contribute to COVID-19 Penetration into the CNS

Since none of the COVID-19 cases with severe neurological complaints reveal evidence of SARS-CoV-2 virus in the cerebrospinal fluid, both direct and indirect mechanisms may participate in CNS dysfunction [20]. The COVID-19 virus infects via the spike glycoprotein (S protein) of the virus binding to the angiotensin converting enzyme 2 (ACE2) receptor on the host cell surface, followed by the incorporation of the viral genome into the host cell genome. ACE2, a cardio-cerebro-vascular protective factor that converts angiotensin (Ang) I to Ang II to stimulate Ang II receptor type1 (AT1R), is expressed by neurons, astrocytes, oligodendrocytes, and endothelial cells of the CNS and is highly concentrated in the olfactory bulb, substantia nigra, middle temporal gyrus, and posterior cingulate gyrus [34]. Activated AT1R can promote inflammation and neurodegeneration and increase blood pressure [35]. Therefore, the widespread presentation of ACE2 as a SARS-CoV-2 receptor on the neural and endothelial cells makes the CNS susceptible to SARS-CoV-2 invasion [36,37]. Since direct access of the SARS-CoV-2 virus is possible through BBB-damaged sites in the brain, its interaction with ACE-2 of endothelial cells in the disrupted BBB provides another penetration route for the virus [38], while transmembrane protease serine 2 (TMPRSS2) facilitates virus entrance through aiding S protein priming. TMPRSS2 inhibitors are beneficial for mental recovery [34]. TMPRSS2 is detectable all over the olfactory epithelium and choroid plexus of rodents and humans [39,40]. In addition to ACE2, basigin or CD147 and neuropilin-1 (NRP1) also operate as docking receptors for the SARS-CoV-2 virus. Furthermore, several proteases like cathepsin B and L (CatB/L) and furin contribute to the viral entrance into cells and its replication [41]. NRP1, a transmembrane receptor without cytosolic protein kinase domain, is highly expressed in the olfactory epithelium and mediates the SARS-CoV-2 invasion into the CNS [42]. As mentioned above, an alternative receptor for the penetration of SARS-CoV-2 into the brain is CD147, which is expressed in neural cells [43]. Additionally, CatB/L, an endosomal cysteine protease, mediates the priming of the SARS-CoV-2 spike protein [44]. It has been suggested that TMPRSS2 and lysosomal cathepsins are able to trigger SARS-CoV-2 invasion; likewise, furin facilitates this process [45].

## 4. Endothelial Cell Infection and Vascular Dysfunction in COVID-19

The vascular endothelia cells are the innermost lining of blood vessels and fundamental for maintaining vessel structural and functional integrity by regulating vascular tone and permeabilization and responses to oxidative stress or inflammation to achieve tissue homeostasis [46].

A recent study displayed the relationship between the levels of endothelia biomarkers and the levels of pro-inflammatory cytokines and chemokines [47]. Along the same vein, recent research indicated that pro-inflammatory mediators released by spike-activated macrophages augment endothelia cells’ activation, likely contributing to the impairment of vascular integrity and development of a pro-coagulative endothelium [48].

Accumulating evidence indicates the presence of extensive microthrombi and endothelia cell damage all over the pulmonary vasculature, suggesting the involvement of vasculopathy in the pathogenesis of COVID-19. Although one study failed to find viral RNA in endothelial cells from COVID-19 patients’ autopsy tissue [49], another study detected SARS-CoV-2 RNA in endothelial cells through in situ hybridization (ISH) in 2 out of 32 cases [47,50]. It has been shown that endothelial cells and pericytes throughout the body express ACE-2 as well as BSG, CD147, and NRP1 [41,51,52,53], supporting the trophism for SARS-CoV-2. Autopsy tissues from COVID-19 patients show high expression of ACE2, TMPRSS2, and endothelial cell inflammation agents in capillaries but less presence in arteries and veins. The existent data suggest that the development of COVID-19 causes endotheliitis in small vessels like capillaries; nevertheless, the involvement of main coronary vessels principally arises from indirect mechanisms of SARS-CoV-2 infection [54].

A recent study utilizing the COVID-19 rhesus macaque model showed that SARS-CoV-2 infection can mediate indirect endothelial cell dysfunction through immune and inflammatory pathways interaction [55]. In another COVID-19 study, the combined activation of the NF-κB signaling cascade and changes in mitochondrial quality control led to endothelia cells’ activation and enhanced neuroinflammation [56]. As a result, SARS-CoV-2-infected human brain microvascular endothelial cells showed augmented caspase 3 cleavage and apoptotic cell death of endothelial cells. 

RNA sequencing data of COVID-19 subjects showed higher expression of the endothelia cell activation markers RELB and TNF-α [57]. Accordingly, it is assumed that hyper-inflammation is responsible for endothelia cell injury and death following SARS-CoV-2 infection. In addition, significantly high levels of Cxcl9 and Cxcl10 in the endothelia cell cluster in SARS-CoV-2-infected K18 mice (SARS-CoV-2-infected human cytokeratin 18 (*K18*)-hACE2) were reported compared to infected WT mice, suggesting interferon pathway upregulation. The Kras pathway was also upregulated. Additionally, upregulation of the Vcam1 and Icam1 genes in the lysosome/apoptosis and complement pathways in viral-RNA-positive cells was evident [58]. A more recent study reported signs of damage and activation of endothelia cells in the form of enhanced gene expression cascades regarding EMT/EndoMT in the hearts of COVID-19-infected patients followed by an upregulation of tissue hypoxia-related pathways and the formation of ultra-structurally detectable thrombi (uTh). Endothelia cells induce the recruitment of monocytes/macrophages to the site of injury through the upregulation of adhesion-molecules and activation of SDF-1/CXCR4 signaling [59]. Overall, these results shed light on the importance of SARS-CoV-2-mediated immune activation in endothelial dysfunction and injury. 

## 5. Degradation of Endothelial Glycocalyx Makes Them Vulnerable to SARS-CoV-2 Entry

The glycocalyx is a layer covering the luminal surface of vascular endothelia cells, which contributes to the maintenance of vascular homeostasis via regulating vascular tone, permeability, thrombosis, and leukocyte adhesion to the endothelium [60]. The glycocalyx is comprised of numerous proteoglycans, glycosaminoglycans, glycoproteins, and associated plasma proteins. The intact highly sulfated glycocalyx structure of the endothelium can prevent SARS-CoV-2 but its disruption increases susceptibility to SARS-CoV-2 infection, resulting in hyperinflammation and oxidative stress [61]. In this regard, a recent study has reported the interaction of intact glycocalyx with the Spike protein and its ability to prohibit the binding of the S protein and ACE2. Conversely, damaged glycocalyx encouraged S protein and ACE2 binding and enabled viral entry [62]. In vivo evidence suggests that COVID-19 patients with severe symptoms showed endothelial glycocalyx disruption and syndecan-1 (SDC-1) secretion (Yamaoka-Tojo 2020). Even among patients who had recovered from COVID-19, the levels of SDC-1 were significantly elevated compared to healthy controls, exhibiting the existence of persistent endothelial damage after COVID-19 progression [63].

## 6. Vascular Inflammation and Blood-Brain Barrier Disruption in COVID-19

Cranial nerves and the BBB are the main avenues between the brain and SARS-CoV-2. However, viruses such as SARS-CoV-2 cannot easily enter the brain parenchyma through the endothelial cells that line the lumen side of the capillaries of the systemic circulatory system due to the unique physiology of the BBB. The BBB is a functional unit that mostly consists of brain endothelia cells linked by TJ proteins, vascular basement membrane, pericytes, astrocytes end processes, neural cells, and microglia [64]. This dynamic structure operates as a bridge between the plasma and brain parenchyma, which are jointly considered as the neurovascular units (NVUs). In the BBB, endothelial cells are fastened by TJ proteins including ZO scaffolding proteins, claudin-5, and occludin as well as junctional adhesion molecules to prevent the extracellular and transcellular diffusion of molecules in the CNS [65]. Hence, in addition to neuroinvasion, disturbance of BBB integrity including endothelial TJ disruption may expose the brain to the danger of SARS-CoV-2 penetration from infected blood, influencing neuronal function in the CNS [66]. Hyper-inflammatory conditions after COVID-19 infection may contribute to BBB damage [67]. In support of this notion, human brain microvascular endothelia cells were reported to overexpress pro-inflammatory cytokines, chemokines, and adhesion molecules and to have lower expression of the TJ protein, which coincided with increased BBB permeability following SARS-CoV-2 infection [68].

Increased pro-inflammatory cytokine levels cause TJ function alteration and disruption of the BBB. For example, IL-1 enhancement damages BBB integrity [69], whereas IL-1β upregulates matrix metalloproteinase (MMP)-9 and promotes TJ protein disruption via activating extracellular signal regulated kinases [70]. Additionally, increased TNF-α, IL-6, and IL-12 cytokine levels result in the deprivation of TJ proteins and consequent BBB permeability impairment [71]. Cytokines likely disturb the integrity of different sorts of junctional proteins such as VE-cadherin, ZO-1, β-catenin, and gap junction, leading to the penetration of inflammatory and immune cells [72]. BBB injury enhances the passing of immune cells through the BBB and increases viral particles and proinflammatory cytokines in the CNS, causing cytokine activation and vascular endothelial growth factor (VEGF) production in astrocytes [73]. By triggering the phosphoinositide 3 (PI3)-kinase and AKT signaling cascades and MMP-9 upregulation, VEGF is capable of disrupting TJ proteins in brain capillary endothelia cells and causing BBB deficiency [74] (Figure 1).

## 7. Disseminating Intravascular Coagulation and BBB Disruption in COVID-19

Both overactive coagulation and hyper-inflammation have been proposed as the main mechanisms underlying endothelia cell damage and thrombus development in COVID-19 [75]. Coagulation is frequently impaired in COVID-19 patients, resulting in a common hypercoagulable state in patients, which may be related to the incidence of stroke. Scientific records imply increased intravascular coagulation, blood clot formation, and bleeding in severe COVID-19 patients. Since thromboembolic single or multiple infarcts are reported in nearly 20% of dementia cases [76], thromboembolic occlusion of cerebral blood vessels is a potential causative of neurological manifestations, including cognitive deficit or dementia, especially in younger healthy adult COVID-19 subjects [77]. As shown in Figure 2, several pathways are involved in intravascular coagulation in COVID-19 patients. (1) The release of pro-inflammatory cytokines can cause platelets’ release, activation, and accumulation. Amongst cytokines, some agents like IL-6 and Cathepsin G, a serine protease generated by neutrophils, have more thrombogenic capacity and can stimulate platelets’ aggregation. In addition, TNF-α induces the surge of plasminogen-activator inhibitor-1 (PAI-1), leading to subsequent decreased activity of plasmin and reduced fibrinolysis [78]. (2) Activation of the complement system is a principal inducer of coagulation. SARS-CoV2 interacts with ACE-2 and activates the complement system including the lectin and classical pathways, resulting in the production of C3a and C3b. C3a mediates inflammation and activates platelets, while C3b contributes to the production of C5a and C5b. The binding of C5a to the C5 receptor mediates the platelet activation, aggregations, discharge of procoagulant microparticles (PMPs), and the development of blood clots. In addition, it may also contribute to the recruitment of neutrophils. C5b generates membrane attack complexes (MACs), serving as a transmembrane channel to initiate the lysis of the embedded cells. The MACs activate the microvascular complement deposition, coagulation, and inflammation. Furthermore, cell lysis and death of target cells are involved in coagulopathy through the enhancement of prothrombin activity as well as von Willebrand factor (VWF) formation [79]. Another mechanism by which complement activation is involved in coagulation is the binding of C3b to the CR1 receptor on the platelet’s membrane. This process triggers the release of short-chain polyphosphate (polyP) from platelets, inducing the expression of tissue factor (TF). (3) Whereas the formation of neutrophil extracellular traps (NETs), composed of chromatin and microbicidal proteins as well as neutrophils, is a key mechanism of conglutination since neutrophils are a crucial player in the production of thromboses, NETs participate in the pathobiology of thrombosis, through which histones, as a main element of NETs, attract and bind to platelets, leading to their aggregation [80]. As indicated in a recent study, neutrophils contribute to the immune response to SARS-CoV-2 invasion. Since these cells are much bigger than erythrocytes and the average capillary diameter, neutrophils can plug capillaries and cause significant blood flow disruption [81]. The adhesion of hyper-activated neutrophils in brain capillaries diminishes the cerebral blood flow in animal models of AD and consequently causes memory dysfunction [82]. Neutrophil-induced disruption of capillary blood flow within the lungs, brain, heart, and other organs is implicated in the poor prognosis of COVID-19 illness [83].

A study utilizing a rat model of intraventricular hemorrhage indicated BBB disruption followed by thrombin-caused activation of Src kinase phosphorylation. Src triggering increases BBB permeability through MMP phosphorylation, TJ protein disruption, and VEGF upregulation [84,85]. In addition, fibrinogen can harm endothelia cells’ integrity by damaging actin filament-attached TJ proteins [86]. On the other hand, enhanced generation of actin likely results in cellular stiffness, actin filament retraction, and spreading of endothelia cells junctions, thus interrupting endothelia cell integrity [87]. In this regard, Yepes et al. discovered vascular leakage in a dose-dependent manner following the intraventricular infusion of endogenous tissue plasminogen activator (tPA) [88] (Figure 2).

## 8. Pneumonia and BBB Disruption in COVID-19

It has been suggested that cerebral hypoxia triggers hypometabolic, cognitive, and degenerative changes in the brain and is contributory to the pathology of Alzheimer’s disease (AD) [89].

A large body of evidence suggests the bidirectional relationship between cognitive outcomes and ARDS, so pneumonia may influence cognitive performance depending on the patient populations and clinical contexts [90]. In contrast, pre-existing cognitive dysfunction is linked to an enhanced rate of mortality by pneumonia [91]. Hypoxemia and hypoxia are considered the causes of neuronal atrophy, consequent enlargement of ventricles, and related cognitive dysfunction, mostly memory loss [92]. In this line, CT scans of patients with ARDS exhibited widespread cerebral atrophy and widening of the bilateral temporal horn compared to the control group [93]. Consistently, pneumonia has been reported to increase the risk of developing dementia among elderly patients [94]. Even non-elderly patients with a single episode of pneumonia without main medical comorbidities are vulnerable to enhanced danger of cognitive decline [94]. According to the report by Wilcox et al., 70 to 100% of ARDS-hospitalized patients indicated cognitive impairment, ranging from difficulties in attention, concentration, and memory to executive function. These changes persisted in 46–78% one year, 25–47% two years, and almost 20% of cases five years after infection [95]. The weakening of memory may result from the susceptibility of hippocampal neurons to oxygen shortage [96]. Therefore, ARDS and hypoxia are vital contributors to the cognitive deficits, especially memory loss, in COVID-19 subjects.

Blood vessels supply oxygen and nutrients to neurons. Continuous hypoxic condition of brain tissue will ultimately cause irreversible neural damage [97]. Postmortem analyses of COVID-19 patients with hypoxic brain damage indicated neuronal injure in the neocortex, hippocampus, and cerebellum areas. Oligodendrocyte demise and widespread gliosis were also reported [98]. A body of evidence demonstrates an important role of hypoxia in BBB disruption [99]. Based on evidence, through changing the actin distribution and attenuating TJ proteins, hypoxia raises the paracellular permeability of brain capillary endothelial cells (Figure 2).

Mitochondria are organelles operating as an energy delivery system. Mitochondrial energy metabolism is directly associated with the hypoxic condition. In addition, hypoxia in the brain may strengthen the proliferative ability of the virus [100]. After hypoxia, a high virus presence in COVID-19 subjects with CNS involvement leads to the compromise of neurons with high-level energy metabolism. Hence, it has been indicated that targeting selective neuronal mitochondrial in SARS-CoV-2 infection induces ‘brain fog’ and causes cognitive and behavioral deficits [8].

In another way, the SARS-CoV-2 virus can result in mitochondrial energy metabolism impairment through targeting oxygen availability and consumption. Notably, the integration of the viral genome into the host cell mitochondrial matrix and creation of a viral-mitochondrial interaction leads to these effects. The interaction between the virus and mitochondria increases energy and reduces host immune reaction, resulting in enhanced replication and survival of the virus [101,102]. Thus, this pathological influence of SARS-CoV-2 infection may elucidate the long-term psychiatric, cognitive, and neurodegenerative outcomes. Furthermore, the reduction in available mitochondrial energy leads to undesired host immune response [103]. Therefore, impaired mitochondrial energy metabolism may be considered as a chief factor in cognitive manifestations in COVID-19 patients.

## 9. Vascular Dysfunction, Brain Inflammation, and Cognitive Impairment

Normally, viruses such as SARS-CoV-2 cannot easily enter the brain parenchyma through the endothelial cells that line inside of the capillaries in the systemic circulatory system due to the unique physiology of the BBB. VE-cadherin, ZO-1, β-catenin, and gap junction caused by the hyper-inflammatory condition of the SARS-CoV-2 virus results in dysfunction and enhanced leakage of the BBB, which exacerbates the penetration of the virus through the disrupted BBB [66]. Once SARS-CoV-2 reaches cerebral tissue, it causes neuro-inflammation through the activation of microglia and macrophages, leading to the release of local pro-inflammatory cytokines in addition to circulatory cytokines. Due to the very high amount of pro-inflammatory cytokines in the circulation, the integrity of the BBB is disrupted, and the brain becomes more vulnerable to ischemic, hypoxic, and thrombolytic threats along with the invasion of various pathogens [104,105] (Figure 2).

Based on research, neurological dysfunction levels and BBB injury seem to be associated with the grade of cognitive loss and of COVID-19 infection severity. A recent study found BBB permeability in 58% of COVID-19 subjects in 31 cases with neurological indications [106], suggesting that SARS-CoV-2 can cause BBB dysfunction. The data of another recent study suggested that the loss of BBB integrity might contribute to the progressive impairment of cognition in diabetic rats. The increase in TNF-α and IL-6 expression might trigger the disruption of the BBB in the brain, which eventually caused cognitive impairment in 8-week-old STZ rats [107]. In support of BBB disruption’s influence on cognitive impairment, a recent work suggested that BBB breakdown takes part in APOE4-connected cognitive weakening independently of AD pathology and might be a therapeutic target in APOE4 carriers [108]. Nevertheless, the accumulation of extracellular β-amyloid (Aβ) plaques and intracellular tau neurofibrillary tangles are considered a main characteristic of the preclinical phase of AD and are important predictors of mild cognitive deficit in cognitively normal persons [109]. SARS-CoV-2 infection is involved in cognitive impairment partially via increased Aβ accumulation. The virus leads to increased neuro-inflammation, changing brain structure, and finally abnormal gathering of neurodegenerative Aβ and phosphorylated tau, resulting in an enhanced risk of cognitive deficiency in patients with COVID-19 [110]. In this regard, a post-mortem investigation of COVID-19 cases indicated significant activation of microglia and neuro-inflammation associated with brain pathology [111]. Microglia, as innate immune cells in the CNS, take part in neurodegeneration pathology. A study showed that microglial distribution and activation in the hippocampus is associated with SARS-CoV-2-virus-induced cognitive impairment [112]. Data revealed that the activation of microglia is a main source of neurotoxic agents, like pro-inflammatory cytokines and reactive oxygen and nitrogen species, causing neural damage progression [113]. It has been illustrated that activating cultured human microglia with neuropeptides causes IL-1β and CXCL8 release [114], which induces pro-inflammatory responses. Microglial-derived pro-inflammatory cytokines and chemokines are able to encourage astrogliosis and Aβ deposition and consequently worsen neuro-inflammation [115].

Likewise, regarding the regulatory role of ACE2 in controlling blood pressure, the occupying of ACE2 by the viral spike protein may result in an imbalance of angiotensin system and affect normal blood pressure. Primary pneumonia and pulmonary infection also cause oxygen deprivation in the brain parenchyma and produce hypoxic conditions and subsequent metabolic disruption [116].

The cytokine storm phenomenon is an increase in pro-inflammatory cytokine levels in the serum, such as IL-2, IL-6 and IL-1β, IL-17, IL-8, G-CSF, GM-CSF, IP10, MCP1, MIP1α (also known as CCL3), and TNF [36], and causes acute respiratory distress syndrome (ARDS) in severe COVID-19 cases [117]. It has been indicated that increased cytokine concentration, especially IL-6, TNFα, and IL-1β, has a strong effect on working memory and attention. Damage of these cognitive abilities is a typical feature of delirium and supports the key role of these cytokines in the etiology of COVID-19-associated cognitive impairments [118].

Prolonged neuro-inflammation and continued hypoxia have been postulated as the potential primary causes of both acute and chronic cognitive features of COVID-19 [105].

Chemokines are small molecules primarily known for regulating the chemoattraction of leukocytes and modulating immune reaction [119]. Additionally, chemokines are involved in different phases of CNS development via supporting cells’ migration, proliferation, and survival [120,121]. Four families of chemokines include C, CC, CXC, and CX3C, characterized based on the conserved cysteine residue position [122]. CCL11 acts through activating the PI3K/AKT, MAPK/p38, and JAK/STAT3 signaling pathways to prevent apoptosis and promote angiogenesis and cell migration [123]. CCL11 also can trigger oxidative stress via microglial NOX1 stimulation and potentiate glutamate-mediated neurotoxicity [124]. During adulthood, the CCL2/CCR2 axis is able to modulate neurotransmission and neuromodulation [123]. It has been indicated that treatment with CCL11 is capable of inhibiting neurogenesis in the adult brain, resulting in cognitive weakening [125].

Enhanced CCL11 concentration was found in plasma samples of long COVID cases suffering from cognitive symptoms, and white matter microglial reactivity was also detected in patients with SARS-CoV-2 infection [126]. In addition, elevated levels of CCL11 are noted in AD and schizophrenia. Accordingly, targeted therapy to normalize CCL11 levels might develop mental and physical health among patients with schizophrenia and AD [127].

A recent study reported white matter microglial reactivity in a mouse model of mild SARS-CoV-2 infection. Furthermore, the same pattern of prominent white-matter-selective microglial responses was found in SARS-CoV-2-infected human brain tissue. In the mouse model, pro-inflammatory CSF CCL11 was elevated at least 7 weeks after infection. In addition, humans suffering from long COVID with cognitive deficits showed higher CCL11 levels than the cases with long COVID without cognitive indications. Likewise, following mild SARS-CoV-2 infection in mice, hippocampal neurogenesis impairment, decreased oligodendrocytes, and myelin loss in subcortical white matter were detected after 1 week of infection, which persevered till at least 7 weeks [128]. The results of this study revealed a crucial role for long-term enhanced CCL11 levels in astrocyte-mediated microglial activation during SARS-CoV-2-induced CNS inflammation, leading to the extended mental and cognitive dysfunction of COVID-19 patients [129]. It has also been suggested that cerebral hypoxia triggers hypometabolic, cognitive, and degenerative changes in the brain and is contributory to the pathology of AD and related cognitive impairment [130].

## 10. Conclusions and Future Directions

In summary, the SARS-CoV-2 virus can invade the brain and exert its neurological manifestation through binding to ACE2 on nerve cells and endothelial cells. A sound body of evidence shows that SARS-CoV-2 impairs vascular integrity through direct or indirect viral infection, leading to endothelium damage and augmenting vascular penetrability in peripheral vessels [131], disrupting the BBB integrity and the CNS function [132].

Given the evidence, the SARS-CoV-2 pathogen can induce cognitive impairment via vascular dysfunction, disruption of the BBB, interruption of oxygen supply, dissemination of intravascular coagulation, and neuro-inflammation. Taken together, the long-term cognitive consequences of SARS-CoV-2 infection, to some extent, may be due to disruption of micro-structural and functional brain vasculatures during COVID-19 illness and in the recovery stages. In addition to the present evidence, future studies are needed to discover the exact long-term cognitive deficits in patients with COVID-19 and their probable mediator mechanisms.

## Figures and Tables

**Figure 1 biology-12-01106-f001:**
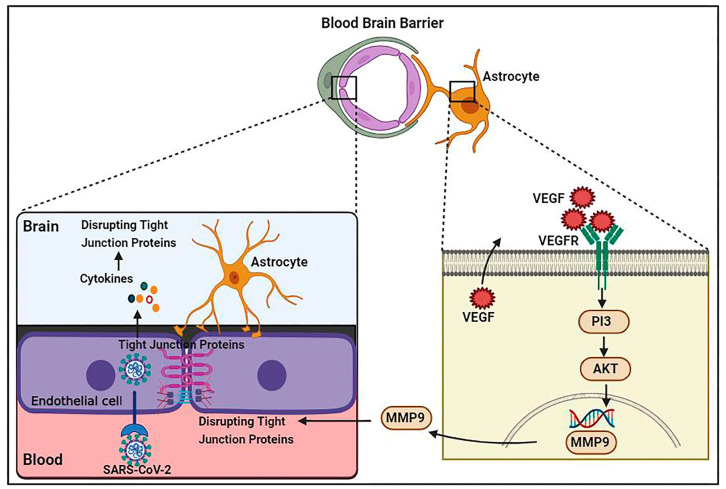
Schematic representation of SARS-CoV-2 infection causing vascular dysfunction and BBB disruption. Proinflammatory cytokines result in VEGF production in astrocytes via allowing immune cells to pass through the BBB. VEGF is able to disrupt TJ proteins in brain capillary endothelial cells by triggering the phosphoinositide 3 (PI3)-kinase and AKT signaling pathways and MMP-9 upregulation, leading to BBB breakdown.

**Figure 2 biology-12-01106-f002:**
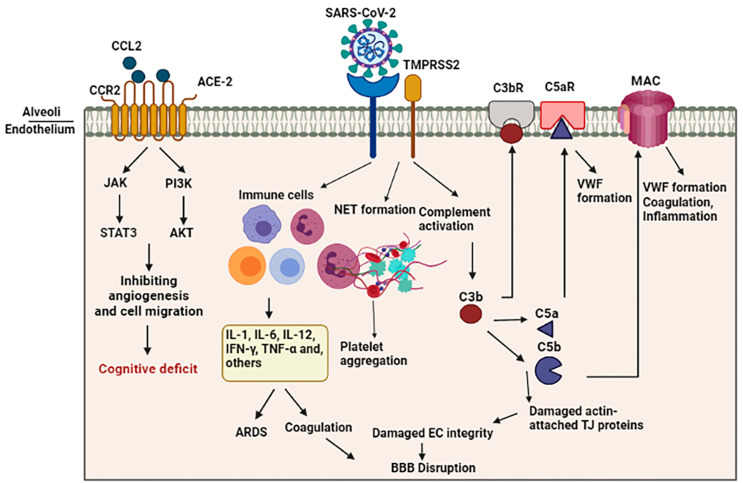
Schematic representation of indirect BBB disruption resulting from vascular dysfunction in COVID-19. The virus acts via binding of the spike glycoprotein of the virus to the angiotensin converting enzyme 2 (ACE2) on the cell surface. Disharmonic and disturbed immune reaction in COVID-19 patients lead to wide local and systemic inflammation by generating cytokine storm, leading to ARDS and vascular coagulation. Alternatively, it may lead to the formation of neutrophil extracellular traps (NETs), composed of chromatin and microbicidal proteins, which participate in the pathobiology of thrombosis and platelets aggregation. Activation of the complement system results in the generation of C5a and C5b, which are involved in the inflammation and activation of platelets. This C5b forms the membrane attack complex (MAC) to activate microvascular coagulation and inflammation.

## Data Availability

Not applicable.

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
