# Peer review of "Vascular Dysfunctions Contribute to the Long-Term Cognitive Deficits Following COVID-19"

_biology, 2023, doi:10.3390/biology12081106_

Round 1

Reviewer 1 Report

This review is complete and well-constructed.

the Author should provide some evidence of other differnt explanation of long-term cognitive deficit, apart from the vascular one (inflammation, glial repair...). This could be added in the "discussion" part, to let the reader be aware of other pathopohisiological explanation.

Author Response

This review is complete and well-constructed.

the Author should provide some evidence of other different explanation of long-term cognitive deficit, apart from the vascular one (inflammation, glial repair...). This could be added in the "discussion" part, to let the reader be aware of other pathopohisiological explanation.

Response: We thank this reviewer for his/her positive comment and for giving us this suggestion. In addition to vascular impairment, neuro-inflammation, synaptic dysfunction, disturbed neurotransmitter release, and neuronal loss contribute to long-term Cognitive deficit [1]. We have added this brief description into the first paragraph of Section 2, “Long-term neurological and cognitive dysfunction following COVID-19”.

Reference

  1. Li, Y., M. Ji, and J. Yang, Current understanding of long-term cognitive impairment after sepsis. Frontiers in Immunology, 2022. 13: p. 855006.

Reviewer 2 Report

Reviewer Comments:

The review manuscript titled “Vascular Dysfunctions Contribute to The Long-Term Cognitive 2 Deficits Following COVID-19” by Zahra Sabani et al concluded that covid-19 affected people has long term neuro cognitive impairments. I appreciate authors for study. I do not have any queries except the following question.

Authors have not mentioned a single point about the amyloid plaques and tau protein post Covid-19infection. Just curious to know that covid 19 has any connection with amyloid plaques and with microglia, a dominant player in the inflammation process.

Author Response

The review manuscript titled “Vascular Dysfunctions Contribute to The Long-Term Cognitive 2 Deficits Following COVID-19” by Zahra Sabani et al concluded that covid-19 affected people has long term neuro cognitive impairments. I appreciate authors for study. I do not have any queries except the following question.

Authors have not mentioned a single point about the amyloid plaques and tau protein post Covid-19 infection. Just curious to know that covid 19 has any connection with amyloid plaques and with microglia, a dominant player in the inflammation process.

Response: We thank this reviewer for his/her positive comment and for raising this question. SARS-CoV-2 infection is involved in cognitive impairment partially via increased Aβ accumulation. The viral infection leads to distribution and activation of microglia in the hippocampus. Activated microglia release neurotoxic agents, like pro-inflammatory cytokines and reactive oxygen and nitrogen species and lead to increased neuro-inflammation, and abnormal accumulation of neurodegenerative Aβ and phosphorylated tau, enhancing the risk of cognitive deficiency in COVID-19 patients [1]. We have added these statements into second paragraph of Section 9, Vascular dysfunction, brain inflammation, and cognitive impairment.

Reference List

  1. Ma, G., et al., SARS-CoV-2 Spike protein S2 subunit modulates γ-secretase and enhances amyloid-β production in COVID-19 neuropathy. Cell Discovery, 2022. 8(1): p. 99.

Reviewer 3 Report

Vascular Dysfunctions Contribute to The Long-Term Cognitive Deficits Following COVID-19

This is an excellent review article discussing recent developments in the long-term cognitive deficits following the COVID-19 pandemic. As we know SARS-CoV-2 is a primarily respiratory pathogen, but it affects other organ systems including the nervous system. This article describes post-COVID-19 long-lasting neurocognitive impairments, which are poorly explored, and how vascular dysfunctions lead to cognitive dysfunctions.  This review article is well-formulated and meticulously written. However, the following are the suggestions to improve the present manuscript further,

1.    In the abstract, the very first line is not good, as the name implies. Please change and rewrite the sentence.

2.    In the first line of the introduction, new variant of coronavirus SARS-CoV-2, but COVID-19, is a disease, not a virus. Please consider rewriting the sentence.

3.    In the introduction lines 44 and 45, there is a repeat of representing (understanding). Please correct this.

4.    Consider revisiting the entire manuscript to better represent the title.

Author Response

This is an excellent review article discussing recent developments in the long-term cognitive deficits following the COVID-19 pandemic. As we know SARS-CoV-2 is a primarily respiratory pathogen, but it affects other organ systems including the nervous system. This article describes post-COVID-19 long-lasting neurocognitive impairments, which are poorly explored, and how vascular dysfunctions lead to cognitive dysfunctions.  This review article is well-formulated and meticulously written. However, the following are the suggestions to improve the present manuscript further,

We thank this reviewer for his/her positive comments and great suggestions.

  1. In the abstract, the very first line is not good, as the name implies. Please change and rewrite the sentence.

Response: We have revised this sentence to “Severe acute respiratory syndrome coronavirus 2 (SARS-CoV-2) is a single-stranded RNA virus and a member of the corona virus family, primarily affecting the upper respiratory system and the lungs.”

  1. In the first line of the introduction, new variant of coronavirus SARS-CoV-2, but COVID-19, is a disease, not a virus. Please consider rewriting the sentence.

Response: We have revised this sentence to “In the late 2019 a new variant of Coronaviruses, SARS-CoV2, emerged as a cause of novel severe acute respiratory syndrome, which had quickly spread to entire world and became the newest global health concern.”

  1. In the introduction lines 44 and 45, there is a repeat of representing (understanding). Please correct this.

Response:  We have deleted the repeat.

  1. Consider revisiting the entire manuscript to better represent the title.

Response: We thank this reviewer for this comment. We have gone through the entire manuscript and made some changes. We hope the revised manuscript is better representing its title.